# Valley Spin–Polarization of MoS_2_ Monolayer Induced by Ferromagnetic Order in an Antiferromagnet

**DOI:** 10.3390/ma17163933

**Published:** 2024-08-08

**Authors:** Chun-Wen Chan, Chia-Yun Hsieh, Fang-Mei Chan, Pin-Jia Huang, Chao-Yao Yang

**Affiliations:** 1Department of Materials Science and Engineering, National Yang Ming Chiao Tung University, Hsinchu 300093, Taiwan; 2Center for Emergent Functional Matter Science, National Yang Ming Chiao Tung University, Hsinchu 300093, Taiwan

**Keywords:** molybdenum disulfide, valleytronics, antiferromagnet, magnetic proximity effect

## Abstract

Transition metal dichalcogenide (TMD) monolayers exhibit unique valleytronics properties due to the dependency of the coupled valley and spin state at the hexagonal corner of the first Brillouin zone. Precisely controlling valley spin-polarization via manipulating the electron population enables its application in valley-based memory or quantum technologies. This study uncovered the uncompensated spins of the antiferromagnetic nickel oxide (NiO) serving as the ferromagnetic (FM) order to induce valley spin-polarization in molybdenum disulfide (MoS_2_) monolayers via the magnetic proximity effect (MPE). Spin-resolved photoluminescence spectroscopy (SR-PL) was employed to observe MoS_2_, where the spin-polarized trions appear to be responsible for the MPE, leading to a valley magnetism. Results indicate that local FM order from the uncompensated surface of NiO could successfully induce significant valley spin-polarization in MoS_2_ with the depolarization temperature approximately at 100 K, which is relatively high compared to the related literature. This study reveals new perspectives in that the precise control over the surface orientation of AFMs serves as a crystallographic switch to activate the MPE and the magnetic sustainability of the trion state is responsible for the observed valley spin-polarization with the increasing temperature, which promotes the potential of AFM materials in the field of exchange-coupled Van der Waals heterostructures.

## 1. Introduction

Transition metal dichalcogenides (TMDs) feature a layered structure similar to graphene but with a direct bandgap, making them ideal semiconductor candidates for various applications [1,2,3]. Structurally, TMDs can adopt three different crystalline forms: 1T (octahedral phase), 2H (trigonal phase), and 3R (rhombohedral phase). Both the 2H and 3R phases exhibit trigonal prismatic coordination and are naturally semiconductors, but the 2H structure is more thermally stable than the 3R structure, making it more suitable for applications requiring stability [4,5,6]. The crystal structure of a TMD monolayer belongs to the hexagonal crystal system, and its reciprocal space exhibits six-fold symmetry [7,8]. When the thickness of TMDs is reduced to a monolayer, the lack of inversion symmetry gives rise to valley degeneracy in the reciprocal space to split into inequivalent K and K’ points with opposite spin states [9,10,11]. This valley spintronic effect makes the electron spin and valley properties interdependent and sensitive to light with angular momentum [7,9,12]. When specific circularly polarized light (either left- or right-handed) is incident on the TMD monolayer, it spin-dependently excites electrons in a specific valley, leading to different occupation levels in the two valleys and resulting in valley spin polarization [13]. This spin-dependent electron transition mechanism, by controlling valley properties, influences the optoelectronic and magnetic properties of TMDs and has given rise to a new research field known as valleytronics. Recent studies have explored various approaches to induce valley spin–polarization, such as using circularly polarized light for photoexcitation [9,14,15] and the Zeeman effect in magnetic fields to break spin symmetry [16,17,18]. However, the aforementioned approaches are volatile in nature, which means the valley spin–polarization would diminish upon the trigger stimuli, such as circularly polarized photons or a magnetic field, being removed. Therefore, those approaches only give rise to the temporary effects on the TMD and limit the feasibility and implementation for TMD-based valleytronics memory. Alternatively, it has been observed that valley spin-polarization can be triggered via the magnetic proximity effect (MPE) of ferromagnetic (FM) materials, which can be achieved at accessible field levels below 1 Tesla, as demonstrated by spin-sensitive spectroscopy [19,20,21]. These findings provide possibilities for realizing TMD-based valley spintronics for more diverse applications. Although the MPE-induced valley spin-polarization in TMD monolayers is clear, an intriguing but unresolved question remains: Can the ferromagnetic order in antiferromagnets enable the MPE in TMD monolayers to incorporate more ingredients into future valleytronics? This issue is promising but requires a more comprehensive understanding.

This study attempts to resolve the magnetic proximity effect (MPE) induced by an antiferromagnet (AFM) nickel oxide (NiO) with a specific crystallographic orientation. NiO is a G-type AFM with a Néel temperature higher than room temperature [22,23]. It has been shown that the (111)-terminated surface of NiO can host the MPE with a depolarization temperature at approximately 100 K, while the (001)-terminated surface of NiO dramatically deactivates the MPE. This phenomenon revisits the importance of the FM order in AFMs through uncompensated spins. This approach provides a new perspective on the study of magnetic Van der Waals heterostructures using AFM materials.

## 2. Materials and Methods

In this study, a three-zone tube furnace was used for chemical vapor deposition (CVD) to grow monolayer MoS_2_ on a sapphire substrate. The molybdenum trioxide (MoO_3_) powders and sulfur (S) powders were used as precursors, positioned in the middle of the second zone and the front end of the first zone, respectively. The sapphire substrates were placed downstream of the MoO_3_ at the second zone. An argon atmosphere with a flow rate of 40 sccm was applied as the carrier gas to promote the reaction of MoO_3_ and S and the subsequent deposition onto sapphire substrates. During the CVD, the pressure inside the tube furnace was maintained at 4 torr. The set temperatures at zones 1, 2, and 3 are 550 °C, 750 °C, and 800 °C, respectively. This creates a temperature gradient inside the tube, promoting the deposition with different distances from MoO_3_ at zone 2. After growing MoS_2_ monolayers on a sapphire substrate, a wet transfer technique was used to transfer the MoS_2_ samples onto the desired NiO substrates with (111)-terminated and (001)-terminated surfaces. Polymethyl methacrylate (PMMA) was spin-coated onto the sapphire substrate with the grown MoS_2_ monolayers. After a soft bake treatment at 120 °C for 2 min, the sample was soaked in a 1M potassium hydroxide solution for 30 min and then immersed in deionized water for cleaning. Due to the hydrophobic nature of PMMA, the PMMA adhering to MoS_2_ would lift off from the sapphire substrate and be transferred onto the surface of NiO with specific surface termination. The desired substrate was then used to pick up the PMMA/MoS_2_ film. After drying, the sample was soaked in acetone for 2 h to remove the PMMA, thus completing the transfer process for subsequent characterizations. The Raman and photoluminescence (PL) spectra were collected using a Raman spectrometer with an argon laser source of 533 nm at room temperature. The laser beam was focused onto the sample’s surface and the spot size was approximately 1 μm. The laser current was 0.755 A, and a 1200-line/mm grating was employed for an energy resolution of 2 meV. The integration time for collecting Raman and PL spectra was 30 and 5 s to yield distinguishable signals, respectively. Spin-resolved PL (SR-PL) was performed based on the pump–probe approach, which utilized linearly polarized light to pump and circularly polarized light to probe the MoS_2_ monolayers on the NiO substrate at various temperatures, which was maintained using a cryostate with liquid helium.

## 3. Results

Figure 1a shows the morphology of the MoS_2_ grown on the sapphire substrate, in which the MoS_2_ domains appear to be randomly distributed and orientated on the sapphire substrate. It reveals the sapphire substrate is atomically flat, thus lacking the preferred nucleation sites on the sapphire surface during the deposition. Figure 1b exhibits the Raman spectrum acquired from the MoS_2_ domain as demonstrated in Figure 1a. The two characteristic peaks at ~384 cm^−1^ ($E_{2g}^{1}$) and ~403 cm^−1^ ($A_{1g}$) correspond to the horizontal and vertical vibration modes of the MoS_2_ monolayer. The monolayer nature of the grown MoS_2_ is indicated by the wave number difference between $E_{2g}^{1}$ and $A_{1g}$, approximately 19 cm^−1^ [24,25]. An atomic-force microscope was employed to probe the step height at the edge of MoS_2_ to examine the properties of the monolayer for the associated valleytronics properties. Figure 1c shows the topography of the MoS_2_ grown on the sapphire substrate, and Figure 1d shows the step profile of the investigated MoS_2_ probed along the scanning trajectory, as shown by the blue line in Figure 1c. As a result, the step height is approximately 0.74 nm, suggesting MoS_2_ in the form of a monolayer fabricated by CVD.

After the fabrication of the MoS_2_ monolayers and the characterization, the focus was turned to the effect of transferring MoS_2_ monolayers onto the NiO substrate. Figure 2a shows the crystal and magnetic structure of the NiO, featuring a rock-salt structure with the AFM texture in a G-type configuration [26,27]. Based on the magneto-structural configuration, the NiO with (111) termination yields a significant FM order on the surface as an uncompensated plane. On the contrary, the NiO with (001) termination is magnetically compensated, resulting in no considerable magnetization on the surface and serving as the natural AFM order in NiO. Figure 2b shows the Raman spectra before and after the MoS_2_ transfer onto the NiO substrate with (111) termination, denoted as NiO_(111)_, in which the $E_{2g}^{1}$ state did not vary notably, but the $A_{1g}$ state appeared to be softened after the transfer. Based on the current literature, the $E_{2g}^{1}$ state is relatively sensitive to the strain/stress issue arising from the different interfacial coherency with the substrate; therefore, the transfer treatment gave rise to limited strain issues. However, the $A_{1g}$ softening suggests the electron doping effect arising from the charge transfer from the substrate [12,28,29,30]. The result suggests that interfacing MoS_2_ with NiO would stabilize the n-type characteristic of MoS_2_. Furthermore, the PL spectra before and after the MoS_2_ transfer both reflect the exciton state of ~1.83 eV as an indicator of the high monolayer quality [31], but it shows the PL redshifting ~10 meV upon transferring the MoS_2_ monolayers onto the NiO_(111)_. Both Raman and PL spectra suggest the MoS_2_ on NiO_(111)_ indeed underwent an electronic transition after the transfer. A detailed discussion regarding the electronic modification on the MPE will be provided after having the results of the SR-PL characterization.

Figure 3a shows the SR-PL spectra of the MoS_2_ on NiO_(111)_ with temperature dependency, probed using right-handed (RCP) and left-handed (LCP) circularly polarized light. The SR-PL taken at 4 K exhibited two distinguishable states at 1.93 eV and 1.97 eV corresponding to the trion and exciton states, denoted as T and A, respectively. The intensity difference between the PL spectra probed by RCP and LCP would reveal the asymmetric spin population between the K and K’ valley, serving as a spectroscopic signature for valley spin-polarization. As a result, the SP-PL taken at 4 K exhibits a distinguishable magnetic circular dichroism (MCD) at both the T and A states, suggesting the valley spin-polarization in the MoS_2_ monolayers triggered by the MPE of NiO_(111)_ substrate. It could be noticed that both T and A states underwent a redshift transition upon increasing the temperature. It is because of the thermal expansion driving electrical band gap reduction, thus reducing the optical band gap [32,33]. In addition, it has been shown both T and A states are responsible for the valley spin-polarization on the LCP and RCP difference while putting the MoS_2_ monolayers on NiO_(111)_. Figure 3b exhibits the SR-PL results of the control sample comprising MoS_2_ monolayers on NiO_(001)_. As expected, the NiO_(001)_ has a fully spin-compensated surface, thus resulting in no distinguishable MCD via the probe of the SR-PL. Comparing the results in Figure 3a,b, it reveals the crystallographic control of the AFM NiO substrate appears to be a critical switch to induce valley spin-polarization via MPE. Figure 3c exhibits the plots of the degrees of the valley spin-polarization, defined by (LCP-RCP)/(LCP + RCP), for the T and A states of the MoS_2_ monolayers on NiO_(111)_ and the A state of the MoS2 monolayers on NiO(001). It shows the depolarization of the T and A states acquired by the MoS_2_ monolayers on NiO_(111)_ is roughly at 100 K, which is relatively high compared to the associated studies [19,34]. Additionally, the valley spin-polarization degree mismatch among the T (NiO_(111)_), T (NiO_(111)_), and the A (NiO_(001)_) at 4 K may reveal two important signatures: (1) The (111)-terminated surface of NiO substrate enables activating the MPE as a crystallographic switch. (2) The T state appears to be much more sensitive to the MPE, hence giving rise to the higher valley spin-polarization degree of ~22%. In order to explain the latter observation, a charge-transfer-induced MPE is introduced, together with the origin of the valley spin-polarization. 

Let us revisit the SR-PL results and discuss the origin of the MPE based on the spin configuration in the MoS_2_/NiO heterostructure. As shown in Figure 3a, both T and A states existed at low temperatures (<100 K) and it should be noticed that the T state is much correlated with magnetism because of the uncompensated spin configuration in the momentum space. Figure 4a conceptually depicts the magnetism associated with the T state, in which the spins carried by the electron and hole located in the *K* valley and the spin carried by the electron in the *K’* valley are exhibited. Consequently, the spins carried by the electron and hole located in the *K* valley would compensate each other in nature, thus resulting in no magnetism. However, the additional spin located in the K’ valley yields the net moment. The T states highlighted by the blue and red triangle shadows represent the local net moments with spin up and spin down, denoted as T(↑) and T(↓), respectively. In general, both T(↑) and T(↓) should populate equally, therefore leading to no magnetism. However, once the MoS_2_ monolayers interface with the NiO with (111) termination, the interfacial exchange coupling between MoS_2_ and NiO_(111)_ would favor one of the T states and induce the imbalanced T population on the net valley spin-polarization. That is, the observed spin-polarized T state magnetically exists and further triggers the valley spin-polarization on the A state, which should be magnetically neutral in nature without interaction with the T state. Therefore, the magnetic sustainability of the T state should also allow the magnetic interaction with other electronic states in the MoS_2_ monolayer, thus boosting the valley spintronic effects toward a relatively high temperature. On the counterpart, upon interfacing MoS_2_ monolayers with NiO_(001)_, the fully compensated surface, as shown in Figure 4b, would result in no population preference on the T state, therefore deactivating the MPE to diminish the valley spin-polarization. This observation suggests precise control over the crystallographic growth of AFM functions as an effective switch to activate the MPE in MoS_2_ monolayers, which is not only scientifically interesting but also practical for valley spintronics.

## 4. Discussion

Although the MPE has been observed on the (111)-terminated NiO surface, there are still two issues regarding the exchange coupling and the depolarization temperature on the T state. Because the uncompensated spins on the NiO surface cannot be easily determined by the external field, the sign of the MPE-induced valley spin-polarization should reveal the information of the uncompensated spins on the NiO. Based on this perspective, the element-specific probe utilizing X-ray magnetic circular dichroism may allow for resolving the exchange coupling at the interface. Therefore, the sign of magnetic dichroism in SR-PL may reciprocally help reconstruct the surface magnetism of the NiO, which is conventionally challenging. In addition, the T state is not as sustainable as the A state upon increasing the temperature. The intervalley scattering on the trion state would be even more profound at the ramped temperature condition, thus leading to the drop in the valley spin-polarization together with the broadening of the SR-PL spectra. The results suggest using a time-resolved SR-PL may further study the MPE-associated properties via the kinetics dimension. Minimizing the scattering on the T state may increase its lifetime and may potentially stabilize the valley spin-polarization at a relatively higher temperature, i.e., >100 K. That is, the ultrafast pump–probe technique should enable studying these MPE-associated issues in the MoS_2_ monolayers for future applications. 

## 5. Conclusions

In this work, the MPE-induced valley spin-polarization in the MoS_2_ monolayers has been uncovered while interfacing the MoS_2_ monolayers with a (111)-terminated NiO substrate studied by utilizing SR-PL spectroscopy. On the (111)-terminated NiO surface, the depolarization temperature of the MoS_2_ monolayers appears to be at around 100 K. By contrast, the (001)-terminated NiO substrate has not been seen with the MPE effect even at the basal temperature of 4 K. The results suggest the precise control over the surface termination for the NiO substrate functions as a crystallographic switch to activate the MPE in the MoS_2_ monolayers adjacent. Electronically, it appears the MPE-induced spin-polarized T state should result from the preferred T population, which naturally carries local magnetism in the specific valley. The valley kinetics of the trions and excitons responsible for the observed MPE should be studied in the next stage to promote MoS_2_-based valley spintronics applications.

## Figures and Tables

**Figure 1 materials-17-03933-f001:**
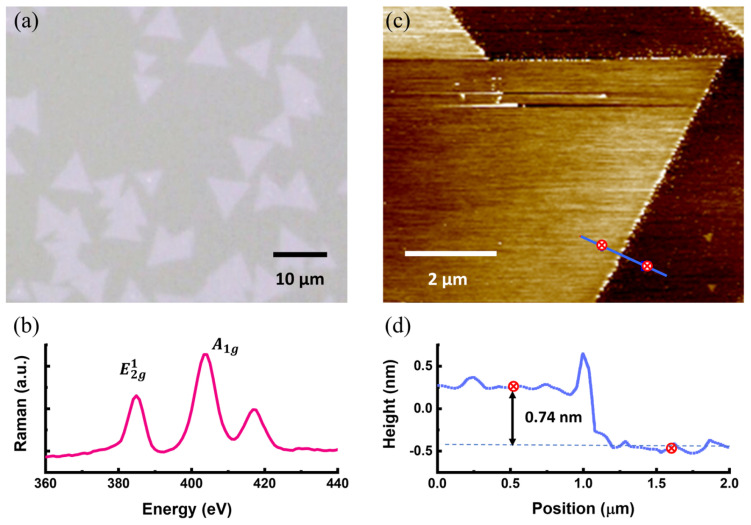
(**a**) Optical microscope image of the MoS_2_ monolayers distributed on the sapphire surface. (**b**) Raman spectrum acquired by the MoS_2_ monolayer in (**a**). (**c**) Topographic image of MoS_2_ monolayer taken by an atomic force microscope together with (**d**) step height of approximately 0.744 nm taken at its edge, as marked by the blue line. The red dots in (**c**) correspond to the red dots in (**d**) to mark the positions for step height scan.

**Figure 2 materials-17-03933-f002:**
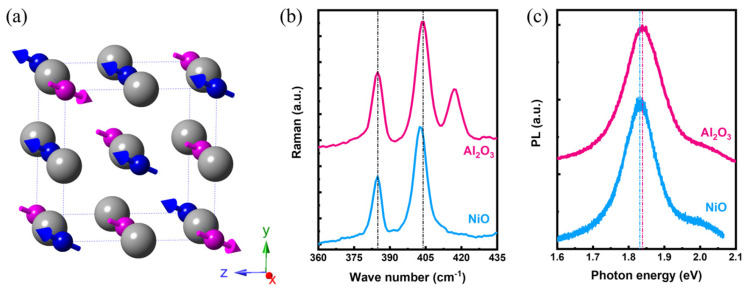
(**a**) Crystal and magnetic structure of NiO substrate exhibiting the rock-salt structure and the G-type AFM texture, respectively, in which the unit cell of rock-salt structure was highlighted by the color and dashed line. The FM and AFM orders of NiO can be observed on the (111)-terminated and (001)-terminated surfaces, corresponding to the uncompensated and compensated plane of NiO. (**b**) Raman spectra and (**c**) PL spectra acquired from the MoS_2_ monolayers on Al_2_O_3_ and NiO_(111)_ substrate.

**Figure 3 materials-17-03933-f003:**
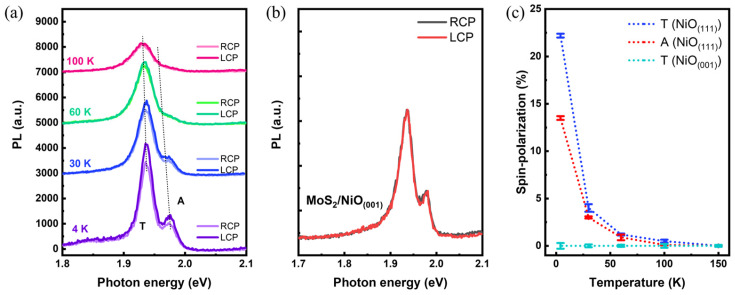
(**a**) SR-PL spectra of MoS_2_ monolayers on NiO(111) taken at various temperatures. (**b**) SR-PL spectra of MoS_2_ monolayers on NiO_(001)_ taken at 4 K. (**c**) Plots of the degrees of valley spin-polarization at T and A states acquired from the MoS_2_ monolayers on NiO_(111)_ and NiO_(001)_ at various temperatures to reveal the depolarization temperature. Temperature during the measurement was precisely controlled using a cryostat with liquid helium.

**Figure 4 materials-17-03933-f004:**
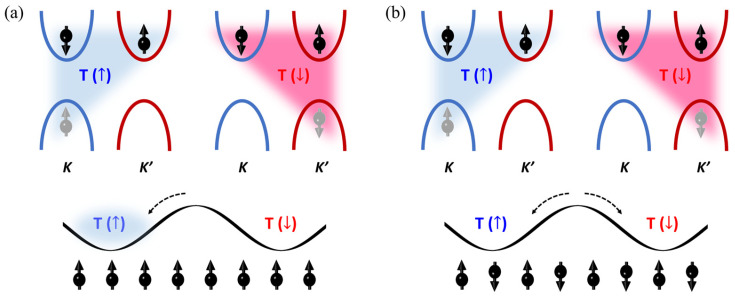
Schematic diagram to demonstrate the spin configuration of trion with (**a**) MPE-induced valley spin-polarization via preferred population to T(↑) driven by (111)-terminated NiO surface and (**b**) neutral population driven by (001)-terminated NiO surface. Arrows at the bottom in (**a**,**b**) present the spin configuration on the (111)- and (001)-terminated NiO surface.

## Data Availability

The authors declare that the main data supporting the findings of this study are available within the article. Extra data are available from the corresponding author upon reasonable request.

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
