# Peer review of "Valley Spin–Polarization of MoS2 Monolayer Induced by Ferromagnetic Order in an Antiferromagnet"

_materials, 2024, doi:10.3390/ma17163933_

Round 1
Reviewer 1 Report
Comments and Suggestions for Authors
In the present work Authors consider experimentally the valley-spin polarization that can be induced in two-dimensional transition metal dichalcogenides (TMDs) monolayers via magnetic proximity effect (MPE). The main motivation is clearly drawn by the Authors through the question stated in the introduction: “Can the ferromagnetic order in antiferromagnets enable the MPE in TMD monolayers to incorporate more ingredients into future valleytronics?”. They attempt to answer this question by resolving MPE induced by an antiferromagnet NiO in MoS2, with a specific crystallographic orientation. As a result, they find that the precise control over the surface termination of the substrate acts as a “crystallographic switch” for activation of the MPE in TMDs.
The subject of the presented paper is timely and worth considering. However, I have doubt about the novelty of this study. First, what new insight does it bring in terms of the MPE and the use of antiferromagnetic substrates? The TMDs where shown before to be sensitive to the antiferromagnetic materials such as MnO (refer to: Phys. Rev. B 97 (2018) 041405(R)). Is the novelty only related to the aforementioned “crystallographic switch” feature or there is something more? In addition, there is no mention why the MEP should be of any interest in the first place? As the Authors state, the valley-spin polarization can be induced via the external magnetic field and related Zeeman effect. What they do not mention is that in this case (the external field/Zeeman effect case) the control over valley-spin polarization appears to be very flexible (refer to: Phys. Rev. B 101 (2020) 115423). In fact, it appears to be much more flexible than in the case of MPE. Hence, what is the reason for MPE? I believe that some answers to this question can be found in mentioned Phys. Rev. B 101 (2020) 115423. Note, that this question should be specifically answered in the context of the main finding i.e. the “crystallographic switch” feature. The Authors should also clearly state that this feature was not reported previously.
From the formal point of view, the study is written in a clear and well-organized manner. I find no major errors except some small inconsistencies e.g. AFM abbreviation is used for both “antiferromagnetic” and “atomic force microscopy”. Several typos/mistakes can be found here and there. In summary, manuscript should be revised to eliminate them.
Author Response
Reviewer 1
In the present work Authors consider experimentally the valley-spin polarization that can be induced in two-dimensional transition metal dichalcogenides (TMDs) monolayers via magnetic proximity effect (MPE). The main motivation is clearly drawn by the Authors through the question stated in the introduction: “Can the ferromagnetic order in antiferromagnets enable the MPE in TMD monolayers to incorporate more ingredients into future valleytronics?”. They attempt to answer this question by resolving MPE induced by an antiferromagnet NiO in MoS2, with a specific crystallographic orientation. As a result, they find that the precise control over the surface termination of the substrate acts as a “crystallographic switch” for activation of the MPE in TMDs.
The subject of the presented paper is timely and worth considering. However, I have doubt about the novelty of this study. First, what new insight does it bring in terms of the MPE and the use of antiferromagnetic substrates? The TMDs where shown before to be sensitive to the antiferromagnetic materials such as MnO (refer to: Phys. Rev. B 97 (2018) 041405(R)). Is the novelty only related to the aforementioned “crystallographic switch” feature or there is something more? In addition, there is no mention why the MEP should be of any interest in the first place?
Response:
First of all, we’d like to appreciate the positive comment in the beginning and the valuable review to let us reconsider the way to deliver the significance of the study.
We agree with the reviewer that the concept of utilizing MPE of AFM is not new. As reported in the literature Phys. Rev. B 97, 041405(R) (2018), it has been shown the MPE can be triggered on the MnO with (111)-terminated surface. However, that work was mainly done by using a first principle calculation. The observation on the MPE of NiO with (111)-terminated surface in our study enables fulfilling the experimental puzzle in this research society. It also demonstrates the feasibility and implementation for the AFM-based valley spintronics using TMDs.
As the Authors state, the valley-spin polarization can be induced via the external magnetic field and related Zeeman effect. What they do not mention is that in this case (the external field/Zeeman effect case) the control over valley-spin polarization appears to be very flexible (refer to: Phys. Rev. B 101 (2020) 115423). In fact, it appears to be much more flexible than in the case of MPE. Hence, what is the reason for MPE?
Response:
We partially agree with the reviewer that utilizing an external field to manipulate the valley spin of TMDs is flexible rather than MPE because of the less stacking issues at the interface. However, in this conventional manner, the field to trigger the observable Zeeman splitting is often at several Tesla, i.e. 7 Tesla [G. Aivazian et al., Nature Physics 11, 148 (2015)], which is not practical for implement. MPE is considered an alternative way to manipulate the valley spin at the accessible field level, often below 1 Tesla. Furthermore, the field-induced valley spin-polarization is so-called “volatile”, which means while the field is removed, the valley spin-polarization diminishes subsequently. On the counterpart, the MPE herein may serve as a non-volatile origin of the valley spin-polarization, which may promote the possible valleytronics-based memory with non-volatility. Together with the current issues concerning the ultrafast switching dynamics and ultra-robust magnetic hardness of AFM, utilizing the AFMs as the MPE host should be able to open a new avenue toward the advanced valley spintronics.
I believe that some answers to this question can be found in mentioned Phys. Rev. B 101 (2020) 115423. Note, that this question should be specifically answered in the context of the main finding i.e. the “crystallographic switch” feature. The Authors should also clearly state that this feature was not reported previously.
Response:
We appreciate the valuable comments from the reviewer. We had added two short paragraphs in the abstract and introduction to highlight the novelty and the potential of utilizing AFMs for MPE, especially via the perspective with a crystallographic switch.
Revision:
(Line 21) This study reveals new perspectives that the precise control over the surface orientation of AFMs serves as a crystallographic switch to activate MPE and the magnetic sustainability of the trion state is responsible for the observed valley spin-polarization with the increasing temperature, which promotes the potential of AFM materials in the field of exchange-coupled van der Waals heterostructures.
(Line 49) …to break spin symmetry [16-18]. However, the aforementioned approaches are volatile in nature, which means the valley spin-polarization would diminish upon the trigger stimuli such as circularly-polarized photon or magnetic field are removed. Therefore, those approaches only gave rise to the temporary effects on the TMD and limited the feasibility and implementation for TMD-based valleytronics memory. Alternatively, it has…
From the formal point of view, the study is written in a clear and well-organized manner. I find no major errors except some small inconsistencies e.g. AFM abbreviation is used for both “antiferromagnetic” and “atomic force microscopy”. Several typos/mistakes can be found here and there. In summary, manuscript should be revised to eliminate them.
Response:
We appreciate the careful review of the reviewer to avoid the potential confusion and misunderstanding. In the revised manuscript, we removed the abbreviation of AFM for the atomic force microscope because it was only used once in the whole manuscript.
Regarding the typos, another reviewer also pointed out this issue and we also revised the content of the manuscript accordingly. We appreciate your great efforts in reviewing our work. We hope the revision has fully addressed all the concerns and the article is at the publishable level of Materials.

Reviewer 2 Report
Comments and Suggestions for Authors
After a careful review, I have found the manuscript to be well-written, thoroughly researched. The results are clearly presented and scientifically sound.
Therefore, I am pleased to recommend the acceptance of this manuscript in its present form.

Author Response
Reviewer 2
According to the authors, this study explores the unique valleytronics properties of transition metal dichalcogenides (TMDs) monolayers, focusing on molybdenum disulfide (MoSâ‚‚). The research investigates how the uncompensated spins of antiferromagnetic nickel oxide (NiO) can induce valley spin-polarization in MoSâ‚‚ monolayers through the magnetic proximity effect (MPE). Using spin-resolved photoluminescence spectroscopy (SR-PL), it was observed that spin-polarized trions are responsible for this effect, resulting in valley magnetism. The findings indicate that the local ferromagnetic order from NiO can significantly induce valley spin-polarization in MoSâ‚‚, with a depolarization temperature of approximately 100 K, higher than reported in related studies. This work opens new avenues for using antiferromagnetic materials in exchange-coupled van der Waals heterostructures, highlighting their potential for valley-based memory and quantum technologies.
After a careful review of the manuscript, I have identified several minor issues that need to be addressed before it can be considered for publication in the journal. I would like to suggest that the authors revise their manuscript in light of the following comments and suggestions before resubmitting it for further review.
Response:
We appreciate the positive comments and the careful review from the reviewer. We would address the concerns raised by the reviewer point-by-point and revise manuscript accordingly. We appreciate your great efforts to help us improve the readability and quality of the paper.
Comment 1: Please clarify that NiO refers to nickel oxide by using its full name (line 14). This improves clarity and helps readers unfamiliar with the abbreviation. The notation of the crystalline forms 1T, 2H and 3R of TMDs should be clarified and properly notated to avoid confusion (line 28-29).
Revision:
(Line 14) This study uncovered the uncompensated spins of the antiferromagnetic nickel oxide (NiO).
(Line 32) Structurally, TMDs can adopt three different crystalline forms: 1T (octahedral phase), 2H (trigonal phase), and 3R (rhombohedral phase).
(Line 61) This study attempts to resolve the magnetic proximity effect (MPE) induced by an antiferromagnet (AFM) nickel oxide (NiO) with a specific crystallographic orientation.
Comment 2: The figure 3 caption contains a typos error “temperatre”.
Revision:
(Line 154) Plots of the degrees of valley spin-polarization at T and A states acquired from the MoS2 monolayers on NiO(111) and NiO(001) at various temperatures to reveal the depolarization temperature.
Comment 3: The statement "It should be noticed that the T state is much correlated with magnetism" is unclear. Please clarify what is meant by the T state being "correlated with magnetism." Provide detailed information on the nature of this correlation and its significance in the context of your research. i.e what they mean by magnetism here.
Revision:
(Line 189) …it should be noticed that the T state is much correlated with magnetism because of the uncompensated spin configuration in the momentum space. Figure 4a conceptually depicts the magnetism associated with the T state, in which the spins carried by the electron and hole located in the K valley and the spin carried by the electron in the K’ valley are exhibited. Consequently, the spins carried by the electron and hole located in the K valley would compensate each other in nature, thus resulting in no magnetism. However, the additional spin located in the K’ valley yields the net moment.
Comment 4: The authors mention that the SP-146 photoluminescence (PL) was taken at 4 K. However, it is not clear how the measurement was achieved at this temperature. Please provide details on the experimental setup or procedures used to achieve and maintain the 4 K temperature for the SP-146 PL measurement.
Revision:
(Line 99) …probe the MoS2 monolayers on the NiO substrate at various temperatures, which was maintained using a cryostat with liquid helium.
(Line 156) Temperature during the measurement was precisely controlled using a cryostat with liquid helium.
Comment 5: Please provide details on the methodology used to measure valley spin-polarization and explain how this measurement is related to the PL spectra. This will help readers understand the connection between the valley spin-polarization data and the PL results presented in the figure 3. The same thing in figure 4.
Revision:
(Line 97) …was performed based on the pump-probe approach, which utilized linearly polarized light to pump…
(Line 161) The intensity difference between the PL spectra probed by RCP and LCP would reveal the asymmetric spin population between K and K’ valley, serving as a spectroscopic signature for valley spin-polarization.
(Line 176) Figure 3c exhibits the plots of the degrees of the valley spin-polarization, defined by (LCP-RCP)/(LCP+RCP), for the T and A states
Comment 6: The authors interpret results based on the presence of trions and excitons. However, there is insufficient explanation on how the results are attributed specifically to the trion and exciton state. Please provide a detailed explanation of how they are considered responsible for the observed results.
Revision:
(Line 170) …it has been shown both T and A states are responsible for the valley spin-polarization on the LCP and RCP difference while putting…
(Line 201) That is, the observed spin-polarized T state magnetically exists and further triggers the valley spin-polarization on A state, which should be magnetically neutral in nature without interaction with T state. Therefore, the magnetic sustainability of T state should also allow the magnetic interaction with other electronic states in MoS2 monolayer, thus boosting the valley spintronic effects toward a relatively high temperature.
Comment 7: The manuscript concludes that "This approach provides a new perspective on the study of magnetic van der Waals heterostructures using AFM materials." However, it is not clear what specific new insights or perspectives this approach offers compared to existing methods.
Revision:
(Line 21) This study reveals new perspectives that the precise control over the surface orientation of AFMs serves as a crystallographic switch to activate MPE and the magnetic sustainability of the trion state is responsible for the observed valley spin-polarization with the increasing temperature, which promotes the potential of AFM materials in the field of exchange-coupled van der Waals heterostructures.
(Line 47) to break spin symmetry [16-18]. However, the aforementioned approaches are volatile in nature, which means the valley spin-polarization would diminish upon the trigger stimuli such as circularly-polarized photon or magnetic field are removed. Therefore, those approaches only gave rise to the temporary effects on the TMD and limited the feasibility and implementation for TMD-based valleytronics memory. Alternatively, it has…
Again, we appreciate your valuable comments and your great efforts in reviewing our paper. We hope all the concerns raised have been completely addressed and the revised manuscript can be acceptable for publication.
